# Fine scale prediction of ecological community composition using a two-step sequential Machine Learning ensemble

Icíar Civantos-Gómez[1,2☯], Javier García-Algarra[3☯]*, David García-Callejas[4,5], Javier Galeano[2], Oscar Godoy[4], Ignasi Bartomeus[5]

**1** Universidad Pontificia Comillas, Faculty of Economics and Business Administration, Madrid, Spain, **2** Complex Systems Group, Universidad Politécnica de Madrid, Madrid, Spain, **3** Department of Engineering, Centro Universitario U-tad, Las Rozas, Spain, **4** Departamento de Biología, Instituto Universitario de Investigación Marina (INMAR), Universidad de Cádiz, Puerto Real, Spain, **5** Estación Biológica de Doñana (EBD-CSIC), Sevilla, Spain

☯ These authors contributed equally to this work.
* javier.algarra@u-tad.com

**Data Availability Statement:** Code and data are available at https://doi.org/10.5281/zenodo.5574457.

## Abstract

Prediction is one of the last frontiers in ecology. Indeed, predicting fine-scale species composition in natural systems is a complex challenge as multiple abiotic and biotic processes operate simultaneously to determine local species abundances. On the one hand, species intrinsic performance and their tolerance limits to different abiotic pressures modulate species abundances. On the other hand, there is growing recognition that species interactions play an equally important role in limiting or promoting such abundances within ecological communities. Here, we present a joint effort between ecologists and data scientists to use data-driven models to predict species abundances using reasonably easy to obtain data. We propose a sequential data-driven modeling approach that in a first step predicts the potential species abundances based on abiotic variables, and in a second step uses these predictions to model the realized abundances once accounting for species competition. Using a curated data set over five years we predict fine-scale species abundances in a highly diverse annual plant community. Our models show a remarkable spatial predictive accuracy using only easy-to-measure variables in the field, yet such predictive power is lost when temporal dynamics are taken into account. This result suggests that predicting future abundances requires longer time series analysis to capture enough variability. In addition, we show that these data-driven models can also suggest how to improve mechanistic models by adding missing variables that affect species performance such as particular soil conditions (e.g. carbonate availability in our case). Robust models for predicting fine-scale species composition informed by the mechanistic understanding of the underlying abiotic and biotic processes can be a pivotal tool for conservation, especially given the human-induced rapid environmental changes we are experiencing. This objective can be achieved by promoting the knowledge gained with classic modelling approaches in ecology and recently developed data-driven models.

**Funding:** O.G. acknowledges support provided by the Ministerio de Ciencia, Innovacion y Universidades (RYC-2017-23666). O.G. and I.B. acknowledge financial support provided by the Secretaria de Estado de Investigacion, Desarrollo e Innovacion (CGL2017-92436-EXP, SIMPLEX and RTI2018-098888-A-I00, MeDiNaS). J.G. acknowledges financial support provided by the Ministerio de Ciencia, Innovacion y Universidades (PGC2018-093854-B-I00). The funders had no role in study design, data collection and analysis, decision to publish, or preparation of the manuscript.

**Competing interests:** The authors have declared that no competing interests exist.

## Author summary

Prediction is challenging but recently developed Machine Learning techniques allow to dramatically improve prediction accuracy in several domains. However, these tools are often of little application in ecology due to the hardship of gathering information on the needed explanatory variables, which often comprise not only physical variables such as temperature or soil nutrients, but also information about the complex network of species interactions that modulate species abundances. Here we present a two-step sequential modelling framework that overcomes these constraints. We first infer potential species abundances by training models just with easily obtained abiotic variables and then use this outcome to fine-tune the prediction of the realized species abundances when taking into account the rest of the predicted species in the community. Overall, our results show a promising way forward for fine scale prediction in ecology.

## Introduction

In the face of human-induced rapid environmental change, the ability to predict species responses to environmental change within a community context is more pressing than ever [1]. However, fine scale prediction is a recognized weak spot in ecology [2–6]. Within the realm of community ecology, most prediction efforts rely on a mechanistic understanding of how multiple abiotic and biotic processes regulate species population dynamics [7]. In particular, theoretical frameworks centered around the study of the determinants of species coexistence and the development of mechanistic models that take into account the effects of the environment and species interactions on the maintenance of biodiversity are an active field of research [8]. These recent developments point out ecological processes that drive the dynamics of interacting species such as those occurring in plant competitive networks [9–11]. Moreover, this body of theory has also shown direct applications to better predict species abundances under controlled experimental conditions [12, 13]. Yet, current theory and associated modelling tools fail in most cases to accurately predict basic features of ecological communities observed in nature such as species abundances, composition, and species turnover in space and time [14]. In order to solve this limitation, there is a recent call to address the complexity of multispecies processes occurring in nature [15, 16]. However, a major stumbling block to advance in this front is parameterizing and validating those models in real communities, which currently is prohibitive due to the complexity of estimating with confidence all parameters from observational data [17]. In order to tackle the problem of the trade-off between model complexity and data availability, we aim to develop an alternative approximation using a mechanistically informed data-driven approach that allows us to achieve predictive power with affordable data requirements.

In a nutshell, existing phenomenological approaches that summarize well-known mechanistic processes require to feed models describing the population dynamics of interacting species with information about 1) the intrinsic ability of species to grow in the absence of interactions, 2) the strength of intra and inter-specific interactions, and 3) how these two sets of parameters change in the presence of different abiotic and biotic variables such as soil conditions or multitrophic species interactions (e.g. pollinators, herbivores) [18, 19]. This is in most cases unfeasible for two reasons: 1) we need to gather detailed information under natural conditions, which for many systems is unfeasible due to the long lifespan of species or the inability to detect and quantify the strength of species interactions, and 2) this approach

considers that all species within a community can potentially interact among them [17, 20]. As the number of parameters to estimate scales exponentially with the number of species in the community, estimating all parameters for large communities quickly becomes an intractable problem. Moreover, because species abundances are not likely to vary independently (i.e. the population size of species A, B, and C covary), it is often difficult to estimate with confidence the strength and sign of many inter-specific parameters. Even if we find a suitable ecosystem to parameterize these models, gathering all required information is labor intensive and highly time consuming. Hence, to resolve this conundrum, we can not rely simply on gathering more and better data. We also need simpler models and search for indirect methods to obtain enough information to be predictive. A key challenge, for example, is that mechanistic models do not always require empirical data that is easy to measure [21]. Hence, we need models that move closer to what we could actually measure on the field. But how to capture complex systems with simpler models?

Fortunately, there is a possibility worth exploring. The problem of inferring key behaviours from complex data has been solved using Machine Learning approaches. Machine learning is a field of computer science that gives computers the ability to learn without being explicitly programmed. In the past decade, Machine Learning has given us self-driving cars, practical speech recognition, effective web search, and a vastly improved understanding of the human genome [22–26]. However its potential has been unleashed mostly in applied domains, as predictions done with Machine Learning approaches often lack the interpretability needed to explain the mechanisms behind the algorithm's decisions. As scientists, we are often uncomfortable with predictions that have no theoretical basis [27]. However, we can combine the power of data-driven models with stronger theoretical foundations [28]. Here we address this issue by partnering together ecologists and data scientists to develop an efficient and predictive data-driven model rooted in known ecological mechanisms that are thought to explain species occurrence and its abundance at local scales. First, we explain the core problem, then we propose a solution, and finally, we test the predictions against a well-resolved data set consisting of five years of observations describing the community composition of 23 species co-occurring in a Mediterranean annual grassland.

## The problem

To predict species abundances within a community context, we know that different abiotic factors determine species performance and their tolerance limits [29], from which one can derive potential species abundances [30]. However, we also know that the final species fate will be modulated by the positive and negative species interactions established among and within species able to grow in a particular place [31, 32]. Of course, stochastic processes coming for instance from dispersal events or random birth and death dynamics [33, 34] are also recognized to have increasing importance in modulating species persistence, but for a first approximation and for the sake of simplicity they are not included in the modelling approach here developed. This is justified as many annual study systems (including ours, see below) complete their life-cycle within a year and "re-start" the next each year. Hence, mechanistic models to understand species population dynamics and their ability to persist in the long-run are often formalized as a set of coupled equations where each response variable (i.e. population size of a given species in a given time and location) depends on and modifies the outcome of the rest of response variables (i.e. population size of this and other species) [31, 32]. A clear example using the standard Lotka-Volterra equations is the persistence of the populations of three plant species following rock-scissors-paper dynamics [35, 36], in which each species have to win and lose simultaneously against different competitors in order to avoid the collapse of the system.

This kind of circular dependence requires measuring all parameters for all species to be able to estimate their behaviour. Even when these parameters are correctly measured spending long hours in the field, the predictive power of such mechanistic models is still very low (See Section 1 in S1 Text).

In our particular scenario, the mechanistic hypothesis is that the abundance of any given plant species is influenced by the environment (e.g. precipitation, soil properties) and the abundance of competitors of that particular season. In mathematical terms, given a subplot $k$ (we drop this index to simplify the notation), the predicted abundance of species $j$ in the subplot $k$ (our spatial sampling unit) at a given season $t$ (year) is:

$$\hat{X}_j(t) = f(A_1(t), .., A_n(t), X_1(t), X_2(t), .., X_m(t)) \qquad (1)$$

where $f$ is a function with $n$ abiotic variables and $m − 1$ abundances of competitors, excluding individuals of $X_j(t)$. Alternatively, it is possible to use data-driven predictive models where the response variable is a function of abiotic and biotic features. While this distinction among features is ecologically important in terms of the ultimate mechanisms driving species abundances, from the point of view of the data scientist that distinction is not relevant, as far as the model behaves properly. The predictive model is just a special class of function:

$$\hat{X}_j(t) = g(A_1(t), .., A_n(t), X_1(t), X_2(t), .., X_m(t)). \qquad (2)$$

Here $g$ is a supervised predictive model, trained with the values of $n$ abiotic variables and $m − 1$ abundances of competitors, excluded $X_j$, from the initial season $t_0$ to $t$. We call it so to make it clear that is does not belong to the set of mechanistic models of Eq 1. Once the model is built, one simply feeds the values of the features at subplot $k$ during season $t$ to predict $\hat{X}_j(t)$. If the available data set includes all these values, the data engineer enjoys a wide range of chances to pick out of them a subset of features to train and tune the model. For instance, if the set includes a long enough series of data recorded during previous seasons $t_0$ to $t$, you can train the model with that set and predict the abundances at subplot $k$ during season $t + 1$, given that all the covariates are available, and that subplot $k$ belongs to the sampled set. That is what we call the *temporal trained* predictive model:

$$\hat{X}_j(t + 1) = g_{temporal}(A_1(t + 1), .., A_n(t + 1), X_1(t + 1), X_2(t + 1), .., X_m(t + 1)) \qquad (3)$$

Note that we do not try to use a time-series approach. You could also use that same model to predict the abundance of $X_j(t)$ at a non sampled subplot $l$. That is the *spatial trained* predictive model:

$$\hat{X'}_j(t) = g_{spatial}(A'_1(t), .., A'_n(t), X'_1(t), X'_2(t), .., X'_m(t)) \qquad (4)$$

where the predictive model is the same as in 3, but the covariates take the values at subplot $l$, season $t$, instead.

While abiotic variables are often easy to measure, obtaining spatially explicit data on species abundances for the whole community is prohibitive, and in fact, it would be equivalent to measuring community composition to predict community composition. If you want to predict the aforementioned abundance $\hat{X'}_j(t)$ you need the values of $X'_1(t), X'_2(t), .., X'_m(t)$.

In any case, and for the sake of being pedagogic, we start by testing the scenario where the full data set is available, and the field team recorded a detailed sample of species abundances and abiotic parameters for each subplot. In this case, it is simple to build a predictive model that works for a nearby piece of land, where all those variables are known: this is the very essence of Machine Learning. So, abundance at $t$ of species $j$ in a given subplot, whose field

data are known but were not used to build the model, could be estimated by Eq 2 by feeding the model with the measured abiotic variables and competitor abundances at that spot. Prediction gets harder when trying to apply the model to a real-world scenario. For example, how do we know in advance the abundance of competitor individuals elsewhere in the community? Eq 4 is deceptive because while the abiotic variables are relatively simpler to measure for the subplot we are interested in, none of the biotic covariates are known at $t$. Similarly, if we want to apply the model to predict the abundances of the incoming season (Eq 3), abiotic features $A_n(t + 1)$ may be gathered without an extraordinary effort, but we would have to wait until we record the number of individuals of each competing species $X_1(t + 1), X_2(t + 1), \ldots$ at subplot $k$. For that task, the predictive model would be less useful.

To put it bluntly, imagine you have sampled 100 areas (i.e.subplots) out of 10000 to build a detailed map of the density of one species that has 20 competitors. The sample is representative of the population and there are no quality issues. Even with that optimal starting point, you would need to count the individuals of competitors species for each of the unknown 9900 plots. The only way to avoid that time-consuming task is to predict those abundances, but as each predictor includes the abundances of competitors the problem is recursive.

A possible strategy to overcome the deadlock is dropping off the conflicting variables. That is, getting rid of the species abundances and relying just on abiotic data, that are easy to measure.

$$\tilde{X}_j(t) = h(A_1(t), .., A_n(t)) \tag{5}$$

This model is valid to predict $\tilde{X}_j(t)$ for an unknown plot at $t$ or for one of the sampled plots at $t + 1$ if we know the values of the set $A_n(t + 1)$. From an ecological perspective this model ignores direct species interactions. For the data scientist, feature engineering is a common procedure to build and test different models. Data sets have redundant information and dimensionality reduction is often desirable. Therefore, we can start building a model with only abiotic predictors. Even from an extreme data-centric approach, this solution looks very weak for this predictive challenge. But *weak* doesn't mean *useless*. A smart mix of weak models may produce an accurate predictor, that is the basis of ensemble methods [37]. This first model generates a set of competing species abundances driven only by abiotic factors. In a second step, we predict again species abundances with the same abiotic data and the predicted abundance of competitors modeled in step one. Thus, we end up with a two-step predictor that is an *ad-hoc* ensemble method for this scenario. The first step, from the abiotic conditions at year $t$, that are easy to measure for each subplot $k$, we predict $\tilde{X}_j(t)$, the abundances of competing species $j$ ignoring the biotic interactions as in Eq 5. The second step, we combine those observed abiotic features with the predicted biotic constrained abundances $\tilde{X}_j(t)$:

$$\hat{X}_j(t) = g(A_1(t), .., A_n(t), \tilde{X}_1(t), \tilde{X}_2(t), .., \tilde{X}_m(t)) \tag{6}$$

where $\tilde{g}$ represents the model of Eq 2. The main difference with the one-step model is that competitor species abundances $X_j(t)$ are replaced by their predicted values $\tilde{X}_j(t)$. This procedure is valid for spatially-trained models (abundance for a subplot $l$ at year $t$) or temporally-trained models (for a recorded supblot $k$ at year $t + 1$) once the abiotic magnitudes have been recorded and choosing the proper training sets.

## Materials and methods

### Data description

We tracked during five years (2015–2019) the local abundances of individuals of 23 annual plant species distributed along 9 plots (plot size 8.5m x 8.5 m) located along a salinity gradient of 1km long by 800 m wide in a highly diverse Mediterranean grassland at Doñana National Park (SW Spain, 37º 04´ N, 6º 18´ W). The plots are placed, on average, > 100m apart from each other. Each plot is subsequently subdivided into 36 subplots of $1m^2$. For each of these subplots, we compiled across each of these five years the number of adult individuals of each plant species at their phenological peak (i.e. when at least half of the individuals are in bloom). This period extends on average and across species from February to June yearly. Thus, overall, we gathered abundance data from 36 subplots in each of the 9 plots, during 5 years, for a total of 1620 plant communities. These subplots represent the basic unit of our study and their scale is appropriate given the small size of the annual plants and the high micro-habitat heterogeneity. For example, the plots in the upper part are rarely flooded, whereas those in the middle and lower parts are annually flooded by vernal pools. This spatial configuration of plots allows capturing small scale variation (due to the different soil conditions created by salinity, among other variables) as well as large scale variation (induced by vernal pools) in the dynamics of annual plant communities in our system. In addition, we empirically measured at the subplot level an array of physical and chemical soil properties at the beginning of the survey (spring 2015) to characterize the abiotic properties of each community (one soil sample per each subplot, see Table B in S1 Text for a summary). These values are kept from year to year for this study as they are stable through time in this type of environments. Finally, we obtained annual precipitation values for each year and the whole study area from a nearby weather station maintained by the regional government "Junta de Andalucía" (El Rocío-Almonte, 10 km far apart).

Soil data was only recorded during 2015 because those are soils with high content of clay in which their properties vary little from one year to another. Therefore, although we acknowledge that within the five years of study some soil properties might have changed, we assume this variation is of little magnitude compared to other abiotic variation such as precipitation and flood.

As initial data assessment, we performed an exploratory analysis, studying abundance distributions for each species and the relationship between their averages and variances. In total, the data set contains abundance values for 37240 species-plot combinations. The distribution of abundances is extremely skewed due to a 75.6% of zero values. Each of them means that the field team has not found any individual of the particular species inside the sampled subplot during the season. Most species are scarcely represented because they were only recorded some years and in some particular plots of the soil salinity gradient. This is a well-known issue in spatial distribution models [38]. Even if zero values were ignored, the uneven distribution of abundances would remain, as generally expected from species-abundance distributions (Fig 1A). The mean value and the variance of abundances scale with each other. This phenomenon is known as Taylor's Law and, in our case, the scaling an exponent of 2.15 and an adjusted $R^2$ = 0.92 [39]. Taylor's Law appears in different contexts in ecology with exponents close to 2 as in this case [40, 41], which implies that our sampling is representative of empirical community structures. In any case, no sample is discarded to build the predictors.

### Methods

**Regression models.**   We implemented in Python three regression models to tackle the problem to predict species abundances (See a full list packages at the end of this document).

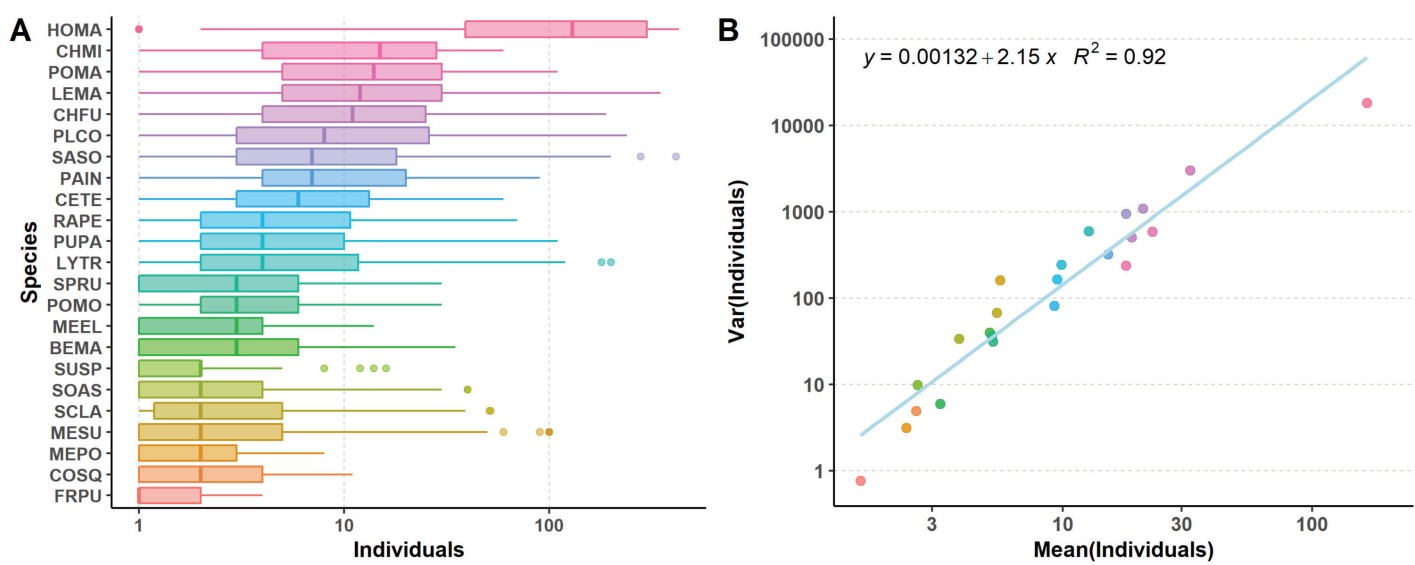

**Fig 1. Species abundances.** A: Boxplots of the distribution of individuals for each species, highlighting the median value. B: Scatter plot of the mean vs. variance for individuals by species, and regression line to check how they fit Taylor's Law.

Linear Regression, Random Forest regression, and XGBoost use the equation presented in 6. Specifically, the Linear Regression Model (LRM) is the simpliest choice to achieve a balance between interpretability and precision. It explains the outcome as a function of the multiple input features and has inspired many mechanistic models. This simple model provides fair results when the underlying function is linear or there are linear combinations of features.

We also used more flexible models to improve results. Random Forest Regression (RFR) is a tree-based ensemble method and belongs to the family of Classification and Regression Trees (CART) [42]. It combines the predictions from multiple weak trees to make accurate predictions [43]. A random subset of samples is drawn with replacement from the training sample. All of them have the same distribution. These randomly selected samples grow decision trees and the average of predictions yields the model's outcome [44]. Alternatively, XGBoost (eXtreme Gradient Boosting) relies on the concept of gradient tree boosting [45, 46]. Boosting is a sequential algorithm that makes predictions for T rounds on the entire training sample and iteratively improves the performance of the boosting algorithm with the information from the prior round's prediction accuracy. It is faster to train and less prone to overfitting than a Boosted Regression Tree (BRT) [47]. XGBoost produces black box models, hard to visualize and tune compared to RFR. Note that our aim is not to compare performance across a wide range of modelling techniques, but to show how different modelling approaches ranging from simple linear regression to to more complex XGBoost can be explored within our framework.

One common feature of all these methods is that they are sensitive to the random splitting of training and testing sets, which we set to an 80/20 ratio. We checked for the spatial autocorrelation of each species abundance and found that for all species the Moran I was low ($I < 0.2$). Hence, we do not further model the spatial component directly in our models, but we do take into account the spatial distribution in our training and testing sets. For each model we perform a 4-fold spatial cross-validation [48] using the K-Folds cross-validator provided by the `Python` package `Verde` [49]. In addition, we provide the results of 100 runs of such models.

**Mechanistic model.** Finally, we show in S1 Text the implementation of a mechanistic model built on a population dynamics framework suited to characterize the dynamics of

annual plant populations [17]. In our implementation, abiotic variables can potentially affect the intrinsic growth rates of the species modelled, as well as intra- and inter-specific interaction coefficients. In S1 Text we show how this model is improved when adding the effect of abiotic variables identified as important by the data-driven model.

**Feature engineering.** The original data set for this regression analysis includes 40 variables. There are 13 abiotic measurements, 12 of soil conditions (pH, total salinity, carbonates, organic matter, C/N ratio, and Cl, C, N, P, Ca, Mg, K, and Na concentrations; Table B in S1 Text) for each subplot, and the annual precipitation, common for all plots. The additional 23 numerical features are the abundances of each species in the subplot (Table C in S1 Text). There is also a factor called *species* that corresponds to the identity of the plant species for which we want to predict its abundance. Note that we build a unique model that works for any focal species, so this factor must be kept to inform the predictor (hereafter we refer to the ABIOTIC and ALLFEATURES datasets in tables and plots).

Decision trees methods, in particular Random Forests, Boost Decision Trees, and Ridge Regression, are not much affected by multi-collinearity [50]. However, since it is a good practice to remove any redundant features from any data set used for training, we used Spearman correlation as a filter-based feature selection method. In addition, for the three models (Linear Regression, Random Forests, and XGBoost), we run a filter feature selection procedure to drop those variables that are less relevant for the outcome [51]. The permutation importance technique tests the performance of a model after removing each feature and replacing it with noise [52].

**Model evaluation.** To assess the performance of regression models we compute the Root Mean Square Error (RMSE) and the coefficient of determination $R^2$ [53]. RMSE is a distance between the vectors of recorded values ($y_i$) and predicted values ($\hat{y}_i$).

$$RMSE = \sqrt{\frac{1}{N} \sum_{i=1}^{n} (y_i - \hat{y}_i)^2} \tag{7}$$

The coefficient of determination $R^2$ is the proportion of of variation of the response variable explained by the regression compared to a null model.

$$R^2 = 1 - \frac{\sum_{i=1}^{n} (y_i - \hat{y}_i)^2}{\sum_{i=1}^{n} (\bar{y} - \hat{y}_i)^2} \tag{8}$$

The second term of Eq 8 is the Relative Squared Error (RSE). It normalizes RMSE by dividing it by the total squared error of the predictor.

**The two-step model.** As we mentioned above, the prediction of abundances in this scenario poses a major challenge as the problem is recursive. To predict the abundance of species *X* we need to know in advance the abundance of each of its competitors, but those abundances are dependent on the rest of the species as well. To solve this limitation and given the fact that soil features and annual rainfall are easier to get, a predictor that could get rid of all abundances is more operative, at the price of reduced predictive power. Dropping that information is equivalent to ignore direct interactions among species. That would be unacceptable for a mechanistic model as a too naïve simplification, but Machine Learning has developed some strategies to deal with this kind of hindrances. Stacked models are a kind of ensemble models that perform sequential learning [54]. Predicted values of stage *n* are fed as features to stage *n* + 1 mixed with original features. We have built a two-step sequential model, following this idea. This stacked generalization predicts the abundances of competing species using the abiotic Random Forest (first step) model and then binds these predicted columns to the abiotic

set to perform the full featured predictor (second step). During the first step, the model is trained with the abiotic data and predicts the abundance of competitor individuals. These predictions may be weak, but combining them with the abiotic variables, we can use this semi-synthetic data set to train an all-features model to perform the final prediction. This can be applied to any modeling tools, and we exemplify it here using Linear Regression, Random Forest, or XGBoost. Specifically, we build 100 models that only differ in the random split of training and testing sets, including all features and years.

In the final step of the analyses, we build full predictors to evaluate spatial prediction by randomly splitting the data in training and testing sets using the spatial cross-validation explained above. When using the model to predict the abundance of a sampled subplot during the incoming season, the training set excludes the samples of the year we want to predict. Please, notice that we may be including years ahead of that predicted (for instance, training with 2018 and 2019 to predict 2017), as our goal is just the evaluation of the goodness of the procedure. We do not explore here other approaches such as the use of time-series data.

## Results

Before building the models we selected the training features by looking at the correlation analysis and Feature Importance. The first method showed two subsets of strongly correlated features (Figs A and B in S1 Text). We kept *C* and dropped *Organic matter*, *N*, and *C/N* ratio. *Salinity* remains in the training set and *Na*, *Cl* and *K* are removed.

After dropping these variables we run the Feature Importance method for the Random Forest with the abiotic set (Table 1). Results show that *Annual precipitation* is the most relevant abiotic feature, after *Species*, that is just the focal species whose abundance we want to predict. *Carbonates*, *C*, *P*, and *Salinity* follow in importance, while *Ca*, *Mg* and *pH* are less relevant than the added random noise, so they could be ignored to build the final model.

We applied the Feature Importance method with the full set of features as well (Table D in S1 Text). Results show that *Annual precipitation* is, again, the most relevant abiotic feature. The number of individuals of abundant competitor species such as *POMA*, *LEMA*, *CHFU*, and *SASO* (see Table C in S1 Text for species acronyms) or the concentration of carbonates showed up to be relevant too for the Random Forest built with the full set.

As a result of both selection procedures, the models (Linear, RF and XGBoost) trained with the abiotic set work with only 6 features: salinity, precipitation, C, Ca, P and carbonates (co3). For the all features and two-step models, we keep Mg and pH as well, because their rank in the importance of features tables was slightly higher for XGBoost. Thus, the training set for these

**Table 1. Feature importance for the Random Forest model with the ABIOTIC set of variables.**

| Feature | Importance |
|---|---|
| *Species* | 0.328 |
| *Annual precipitation* | 0.250 |
| *Carbonates* | 0.098 |
| *C* | 0.091 |
| *P* | 0.067 |
| *Salinity* | 0.049 |
| *Random noise* | 0.036 |
| *Ca* | 0.028 |
| *pH* | 0.027 |
| *Mg* | 0.023 |

two latter models includes 8 abiotic and 23 biotic features, one for the abundance of each species.

As the model is unique, there is a circular problem with the abundance of individuals of the focal species when acting as competitors. For instance, to predict the abundance of *HOMA* individuals in a particular subplot, we should know in advance the abundance of *HOMA* individuals as competitors. Getting rid of the *HOMA* column is unfeasible, because those values are important to predict the abundance of any other species. So, before building the full set model, this value is set to 0 where the competitor and focal species are the same. We keep the rule for the two-step predictor.

We found that models that include the full set of abiotic and biotic features perform quite well regarding their $R^2$ to predict species abundances within a spatial context. This is an important result because it shows a direct application of using Machine Learning approaches to describe relevant characteristic of ecological communities such as the spatial distribution of species relative abundances. Specifically, we build 100 models that only differ in the random split of training and testing sets, including all features and years. The median $R^2$ values are 0.095 for the Linear Regressor, 0.809 for XGBoost and 0.867 for Random Forest (Table 2). Prediction would be a practical tool with these two former models, but results may be deceiving to ecologists. To predict the abundance of species *X* we need to know beforehand the abundances of the rest of species, so the painstaking field work is not avoided.

The weak performance of the Linear Regressor is a hint on the non-linear nature of the prediction challenge. The F statistic for the abiotic data set trained LRM, is nearly null. According to the *t* value, the order of significance of variables is *Ca*, *C* and salinity, with the annual precipitation in fourth place (Table E in S1 Text). Even though is a rough way to compare, the Feature Importance for the RFR model is quite different, with the annual precipitation as the most important variable (Table 1).

The median $R^2$ value for the Random Forest predictor trained just with abiotic information is very close to the predictor trained with all features: 0.852 vs. 0.867. This figure provided the hint to try the two-step method. Results are quite encouraging as the median $R^2$ of two-step models is 0.868 using Random Forest for the second stage and to 0.831 using XGBoost. The median $R^2$ of the two-step is virtually identical to the value 0.867 we got with the model built with the full data set. The same happens when we compare the median *RMSE* values of both methods: 14.290 (two-step) vs. 14.361 (all features). The practical advantage of the two-step method is that it does not require to know in advance the abundance of competitor species.

Fig 2 shows the improvement of the $R^2$ distribution with the two-step method and Random Forest as the second stage model. XGboost results were slightly worse (Figs C and D in S1 Text).

Although $R^2$ is useful to make global comparisons among predictors (i.e. among species), we still require an assessment of prediction accuracy by species because of their asymmetry in observed abundances. To evaluate the three methods considering a species-specific approach, we performed 100 runs, following the steps described in the previous section, and measured both RMSE and RSE for each species ($RSE = 1 - R^2$, just for plotting convenience using a logarithmic scale). We overall found that relative squared error is fairly small for abundant species such as *Hordeum marinum* or *Chamaemelum fuscatum*, while it shows a wide spread for plants that are relatively rare in the study area (Fig 3, see also Figs E and F in S1 Text).

Fig 4 shows the distribution of errors of a particular run. The two-step Random Forest model seems to be much more accurate predicting zeros than the abiotic RF model.

The Random Forest models do not predict negative values. The Linear Regressor and XGBoost, return between 9% and 25% of negative values that would not have biological sense.

**Table 2. Prediction errors for spatial application.**

| Model | Median $R^2$ | | | Median RMSE | | |
|---|---|---|---|---|---|---|
| | Linear | Random Forest | XGBoost | Linear | Random Forest | XGBoost |
| All features | 0.095 | 0.867 | 0.809 | 37.505 | 14.361 | 17.234 |
| Abiotic features | 0.024 | 0.852 | 0.827 | 38.969 | 15.138 | 16.383 |
| Two-step model | 0.222 | 0.868 | 0.809 | 34.789 | 14.290 | 16.171 |

We have kept them as predicted to compute the $R^2$ index, and we have kept the decimal values as well, in order to make fair comparisons among the different models.

Similarly to predicting species abundances across space, we could predict species abundance over time with the same models trained with a different data set. From a modelling perspective, prediction over time is a widespread application of Machine Learning. If we have got a curated yearly series of data, it is straightforward to build a predictor for the incoming season, and in the case the quality of predictions is fine enough, then it would allow us to anticipate how plant will respond to changes in future environmental conditions.

Unfortunately, this expectation is not the case for the data analyzed, and it comes as no surprise. This annual plant system is a highly variable system in which propagules can disperse over a wide range of distances after individuals complete their life cycle. Such dispersal kernels in combination with variation in flooding events make our system overall highly dynamical in terms of space and time. We therefore do not believe that our system is stable through time as it can be other systems with species with longer life cycles such as shrubs or trees Table 3 shows the evaluation results of temporal trained predictors, including all features of four years and tested with the remaining one. The median $R^2$ values are very disappointing for all models. A potential explanation is that, despite the fair size of the data set, the temporal sample is tiny. In addition, yearly fluctuations in weather are heavily marked in this study system, ranging from 384 mm in 2019 to 625 mm in 2016. The fact that there are only five values for time-related variables, one per year, makes prediction to fail because the test data often falls outside the trained data conditions. One possible workaround is dropping the annual precipitation to reduce overfitting, but having in mind that feature analysis showed that it is the most relevant independent feature. Results show a mild improvement but even the best $R^2$ (0.08 for 2019) tell us that the predictive value is nearly null. Results for Linear Regression and XGBoost predictors are even worse.

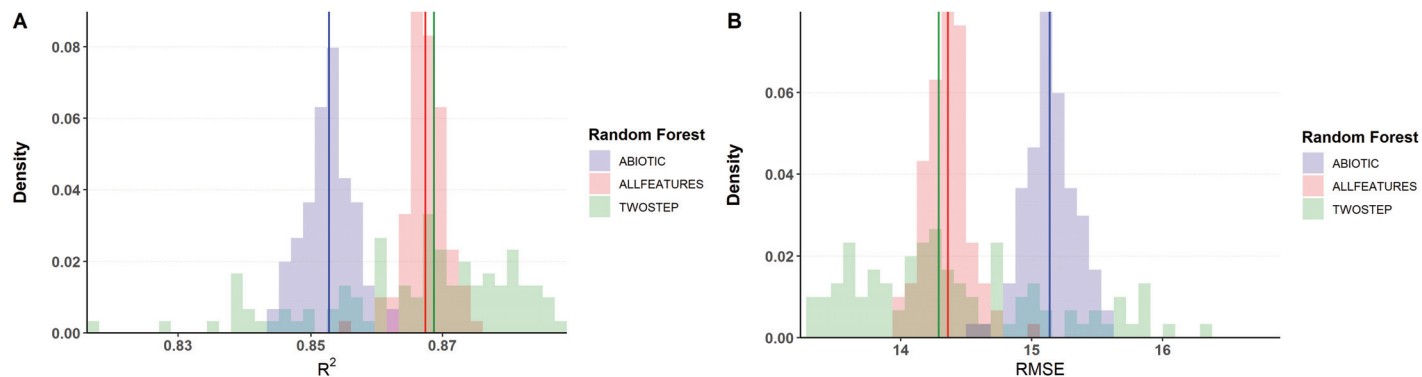

**Fig 2. Prediction errors with a two-step Random Forest Regressor.** A: Relative Squared Error distributions for 100 random choices of training/testing sets, vertical lines set at median values. B: Root Mean Square Error distributions for the same collection of predictors.

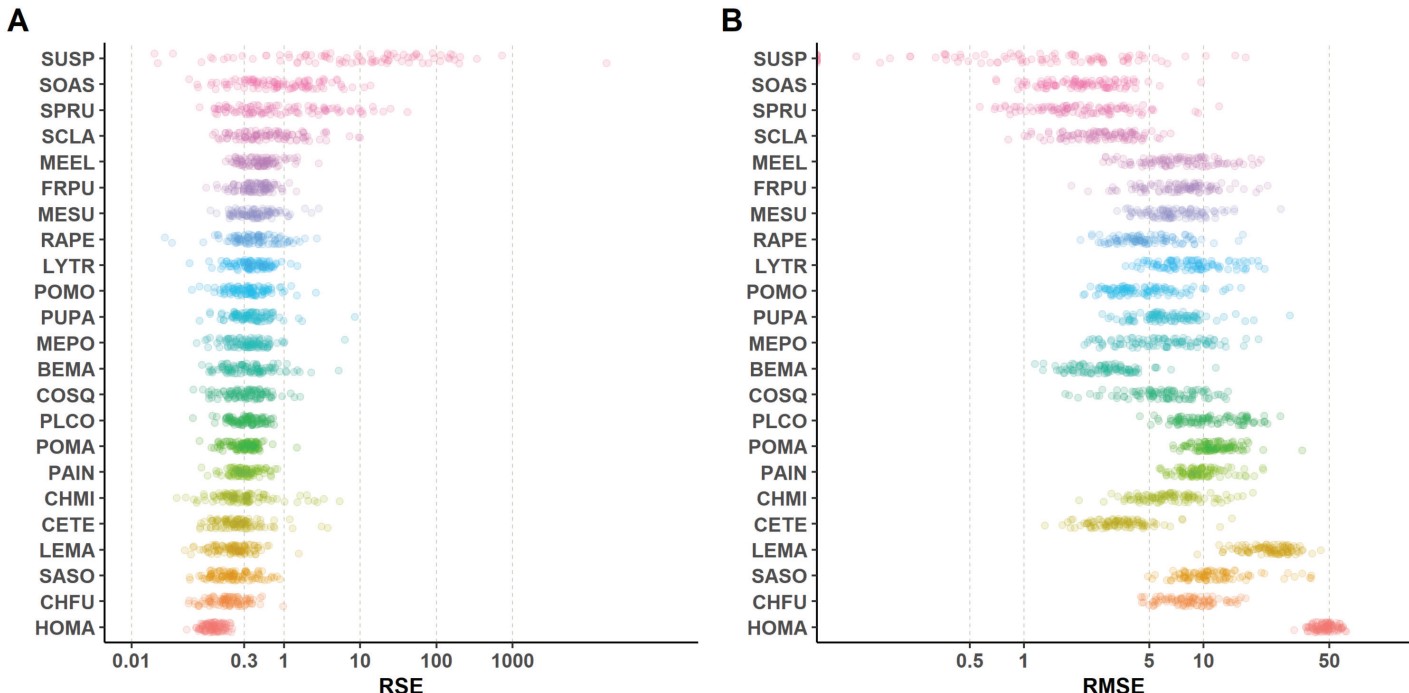

**Fig 3. Prediction errors by species using a two-step Random Forest Regressor.** A: Relative Squared Error distributions for 100 random choices of training/testing sets. B: Root Mean Square Error distributions for the same collection of predictors. See Table C in S1 Text for species acronyms.

Regardless of the differences in the ability of the Random Forest models to predict species abundance over time or across space, these models have the potential to provide novel insights into some key processes that modulate the response variable studied (species abundances in our case). This new information can be incorporated in turn into mechanistic predictions from population dynamics models that describe the abundance trajectories of interacting species. These later type of models are much more familiar to ecologists. This possibility of feedback from the data-driven models to the mechanistic models is exemplified in our system with the particular focus on soil carbonates. The inclusion of this abiotic variable, which was deemed second in importance just after the annual precipitation by the Feature Importance method (Table 1), shows an overall improvement in the predictions derived from the mechanistic models (S1 Text).

## Discussion

By combining ecological knowledge with data-driven models, we showed that it is possible to develop reliable models that predict reasonably well complex systems such as the abundance of multiple species that compose ecological communities. Plant species composition at fine-resolution scales is hard to predict, because their densities and relative abundances are partly governed both by abiotic factors, which determine where species can potentially thrive, and by the network of species interactions in which they are embedded, which modify their reproductive success. In fact, these two axes of variation defining the species persistence probabilities have been at the core of the species niche concept [55], and in the development of modern community ecology theory [56], but rarely exploited for predictive purposes. Here, we show a simple methodology to use easy to obtain abiotic information to accurately predict species abundances while taking also into account their potential biotic interactions. Our models are

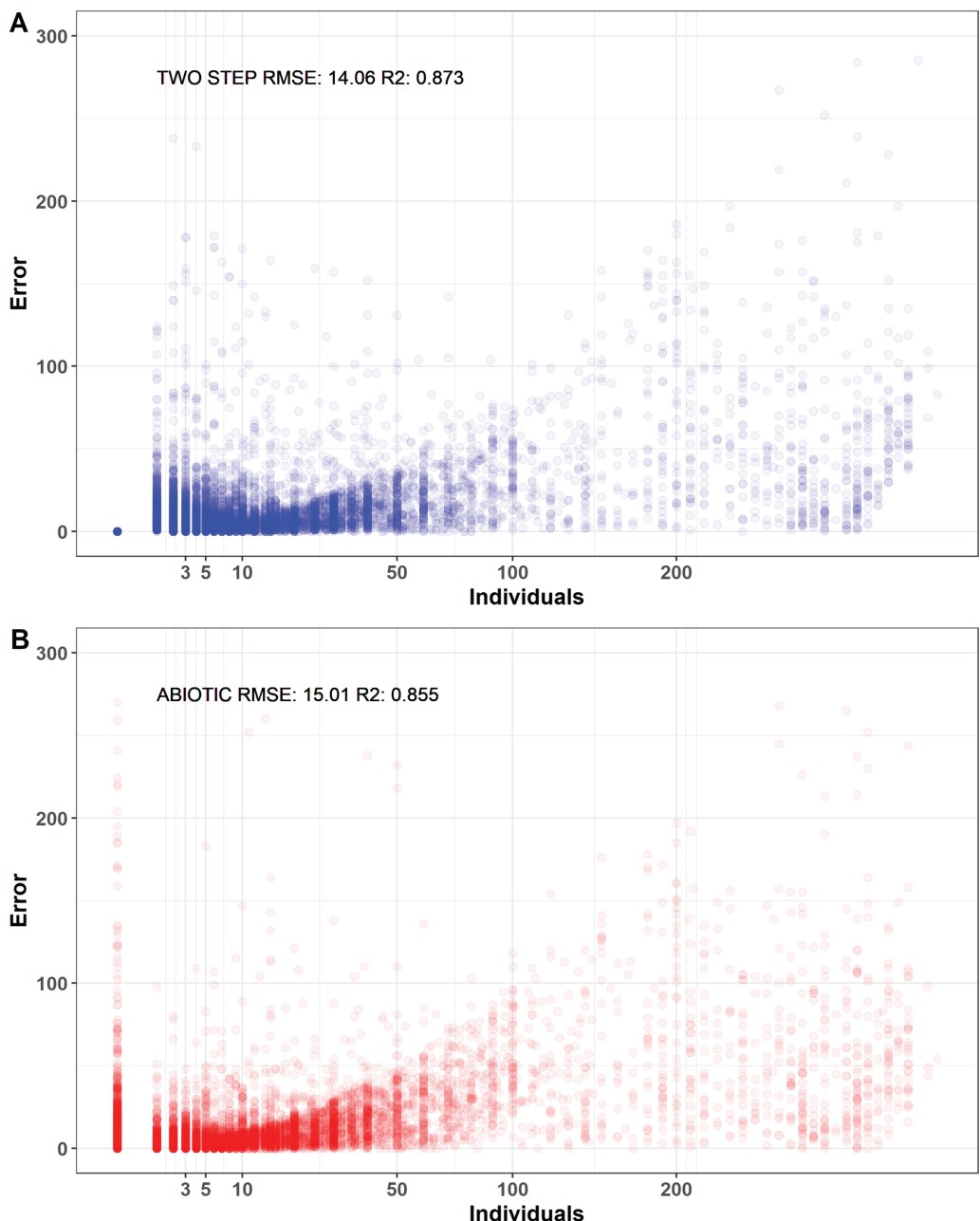

**Fig 4. Prediction errors by individuals.** Each dot is the value of $|y - \hat{y}|$ where $y$ is the recorded value of abundance and $\hat{y}$ the regression prediction. There are 37260 predictions for each run. A: Error values for a run of the two-step model with Random Forest. B: Error values for a run of the abiotic model with Random Forest.

sensitive to the breadth of the training data, and as such they capture better the spatial anomalies (where we have more data) than the temporal anomalies. For this last practical purpose an alternative approach based on time-series may yield better results.

Machine learning-based methods have been extensively applied for relating species distributions to environmental factors, through species distribution models. While the literature on species distribution modelling is vast, most of it is centered on large scale distributional patterns of species occurrences [57], often involving only abiotic variables [58], and in a vast

**Table 3. Prediction errors splitting by year and using Random Forest.**

| Predicted Year | With Precipitation | | Without Precipitation | |
|---|---|---|---|---|
| | Median RMSE | Median $R^2$ | Median RMSE | Median $R^2$ |
| 2015 | 33.39 | -3.95 | 27.77 | -2.42 |
| 2016 | 21.58 | -1.34 | 21.77 | -1.38 |
| 2017 | 49.54 | -0.01 | 47.26 | 0.08 |
| 2018 | 42.74 | -0.16 | 41.47 | -0.09 |
| 2019 | 54.81 | 0.07 | 54.85 | 0.07 |

majority of cases, prediction is limited to species presence or absence. However, most ecosystem functioning processes happens at the community scale. At this scale, species interactions are thought to determine species performance, quantified in their probability of persistence [31] and in their relative abundance [59]. We show that a data-driven sequential model that firstly predicts the potential species abundances for a given set of abiotic variables, and secondly uses this predictions to refine the realized species abundances predicted, performs fairly well when comparing them with more data-hungry models. However, note than when the species abundance is low (median value under 3 individuals), the uncertainty of the abiotic prediction increases. To avoid this issue, the model could be refined in a future development through careful resampling of low abundance species before performing the first step. We discarded this procedure because to raise the overall $R^2$ with simple SMOTE-based resampling required too high resampling percentages for this particular application [60]. In any case, a remarkable fact is that the two-step model is much better predicting at absences than the abiotic one. The existence of competing species seems to play an important role as an inhibitor of the growth of a particular species. This information is lost when the model only works with abiotic features. The fact that this two-step process matches the predictions of a one-step model with all data available is remarkable. One possible explanation is that observed plant abundances empirically measured in the field only capture fully developed individuals, missing early stages of competition among seedlings that despite dying soon, affect final species abundances.

In our case, the best performing data-driven model is the Random Forest, closely followed by XGboost. It was expected that the assumptions of linear models are too simple when there are complex interactions among features, as the exploratory analysis suggested. Which model is more appropriate may depend on the data set at hand. Interestingly, this data-driven exercise can also help us enhance mechanistic models. We already used mechanistic models to understand the species dynamics in our ecological system. Aware of the importance of the abiotic environment, we modelled species reproductive success as a function not only of competitors, but also of other environmental variables such as soil salinity content [18]. To our surprise, the feature importance selection procedure highlights $CaCO3$ as a key determinant of species abundances and not salinity, which was the most obvious variable initially selected in the field. Despite initially counter-intuitive, this result is congruent with the fact that we sampled in a hypersaline environment in which phosphorous (a key element for plant growth) is not available for plant absorption. Rather, it is retained in carbonate minerals such as calcite and dolomite, and plants can mostly obtain phosphorous thanks to the enzymes from mycorrhizal fungi. With this new knowledge, we re-parameterized the mechanistic annual plant model by adding $CaCO3$ as a covariable affecting both the intrinsic fecundity rates and the pairwise interactions among species. With this update we obtained significantly better predictive error than with the biotic-only parameterization (Table A in S1 Text). Hence, we show that ecological process can shed light on data driven models, but those can in turn refine

which ecological process are important to include in the mechanistic models. In our relatively simple proof of concept, the mechanistic formulation of the parametric model was not influenced by the data-driven model, but more complex feedbacks are of course conceivable, for example more appropriate functional responses (e.g. non-linearities) of some variables, or the interaction among variables. In any case, data-driven methodologies are specially suitable when one has data on many different environmental variables, which would be unfeasible to include in a parametric model one by one.

This exercise is tailored to the problem at hand. For example, an implicit assumption of this modelling framework is that plant species can reach all quadrants in the grassland, and are not limited by dispersal. This assumption is reasonable on a study system in which seeds are small, they can be dispersed by wind and small animals such as ants, and additionally the system also gets flooded in extremely wet years. Similarly, we focused our modelling on the plant-plant competitive interactions, which are the main interactions structuring this grassland communities [61], and ignore other interactions such as pollination or herbivory. However, the same approach can be used to model other interaction types in other systems, as far as you have initial data to train the models. However, when modelling species with lower detectability than plants or hyper-diverse communities, further enhancements may be needed to obtain sensitive results. In our case, we obtain a good spatial predictive ability, but we fail to predict temporally. Given the strong across-year variations in precipitation, we believe this is due to the limited number of years to train the data, and not an inherent limitation of the framework. It might also be possible that stochastic events, which create variation from unknown sources (e.g. random birth-death, perturbations in population sizes, dispersal events in no particular direction) are more prevalent in the temporal dimension than deterministic processes such as species interactions [62]. In any case, given the expected ongoing environmental change in many abiotic variables such as precipitation regimes and temperatures, we envision this kind of predictive models to be specially suitable in combination with semi-automated species monitoring schemes (e.g. NEON, [63]) to anticipate to global change effects on delicate and highly-diverse ecosystems such as Mediterranean grasslands. We want to highlight that the proposed approach complements current approaches to understand fine scale community composition, such as multivariate methods (e.g. CCA [64]) or time series analysis [65], which may be more suitable depending on the question to be answered, or the data available. Including the temporal resolution of soil properties may enhance model performance.

## Conclusion

The rate of ecological data generated is increasing substantially [63]. Open and reliable data sets hold the potential to facilitate the application of near-term forecasting protocols [6]. However, for those efforts to thrive, we need simple models that can work with the sparse data typical of ecological surveys. A more predictive ecology likely serves to anticipate how several ongoing critical environmental changes such as climate change affect multiple properties of ecosystems, and at the same time it also provides information about which management actions are required to maintain healthy ecosystems. Taken together, our results show that two-step ensemble models are a promising tool to reach efficient management without the costs of prohibiting data collection.

## List of packages

```
Python: python 3.8.8 [66], matplotlib 3.3.4 [67], numpy 1.20.1 [68], pan-
das 1.2.4 [69], seaborn 1.11.1 [70], scikit-learn 0.24.1 [71], verde
1.6.1 [49], xlsxwriter 1.3.8 [72], xgboost 1.4.2 [73].
```

```
R: r-base 4.1.0 [74], cowplot 1.1.1, [75], ggplot2 3.3.3 [76], gridExtra
2.3.0 [77], patchwork 1.1.1 [78], scales 1.1.1 [78], tidyverse 1.3.1 [79].
```

## Supporting information

**S1 Text. Abundance prediction with population dynamics models and supplementary figures and tables.**
(PDF)

## Acknowledgments

We thank Doñana National Park staff for granting access to Caracoles real estate.

## Author Contributions

**Conceptualization:** Javier García-Algarra, Oscar Godoy, Ignasi Bartomeus.

**Data curation:** David García-Callejas, Oscar Godoy, Ignasi Bartomeus.

**Formal analysis:** Icíar Civantos-Gómez, David García-Callejas, Oscar Godoy, Ignasi Bartomeus.

**Funding acquisition:** Javier Galeano, Oscar Godoy, Ignasi Bartomeus.

**Investigation:** Icíar Civantos-Gómez, Javier García-Algarra, Javier Galeano, Oscar Godoy, Ignasi Bartomeus.

**Methodology:** Icíar Civantos-Gómez, Javier García-Algarra, Oscar Godoy.

**Software:** Icíar Civantos-Gómez, Javier García-Algarra.

**Visualization:** Javier García-Algarra.

**Writing – original draft:** Icíar Civantos-Gómez, Javier García-Algarra, Ignasi Bartomeus.

**Writing – review & editing:** Icíar Civantos-Gómez, Javier García-Algarra, David García-Callejas, Javier Galeano, Oscar Godoy, Ignasi Bartomeus.

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
