## [Decision Letter · Decision Letter 0]

9 Jun 2021

Dear Dr García-Algarra,

Thank you very much for submitting your manuscript "Fine scale prediction of ecological community composition using a two-step sequential machine learning ensemble" for consideration at PLOS Computational Biology.

As with all papers reviewed by the journal, your manuscript was reviewed by members of the editorial board and by several independent reviewers. In light of the reviews (below this email), we would like to invite the resubmission of a significantly-revised version that takes into account the reviewers' comments.

Reviewer 1 and 2 raise important methodological concerns which need to be addressed. In particular, as also suggested by reviewer 2, I encourage the authors to better and more deeply discuss the limitations of their approach.

We cannot make any decision about publication until we have seen the revised manuscript and your response to the reviewers' comments. Your revised manuscript is also likely to be sent to reviewers for further evaluation.

Sincerely,

Jacopo Grilli

Associate Editor

PLOS Computational Biology

Natalia Komarova

Deputy Editor

PLOS Computational Biology

Reviewer 1 and 2 raise important methodological concerns which needs to be addressed. In particular, as also suggested by reviewer 2, I encourage the authors to better and more deeply discuss the limitations of their approach.

Reviewer's Responses to Questions

**Comments to the Authors:**

Reviewer #1: uploaded as an attachement

Reviewer #2: Summary:

This paper proposes a two-step ensemble model and feature engineering workflow to predict species abundances in real world populations over space and time. The aim is ambitious, and it speaks to an important division in quantitative ecology between traditional mechanistic population models, mostly using formal Bayesian models with MCMC, and more flexible machine learning models. The work is similar to semi-supervised learning with weak labels where the predictions from the first step are propagated to avoid measuring difficult interaction covariates in the field. They provide a nice worked example. I like the idea and think the conversational tone throughout is useful. Given that formal models have been largely non-effective, and that there is no hope is directly measuring species interaction coefficients over long periods of time, we definitely need all new ideas we can get. To make this paper more palatable to a large audience, the introduction needs to slow down, avoid making grand statements, and really consider the current state of quantitative ecology. My only concern for the results is that with a single example, and no simulations, it is difficult to guess which parts of the model will be most sensitive to typically challenging ecological scenarios. A short list might include

• Incomplete detection

• Large clines in environmental conditions

• Spatial autocorrelation

• Rare events/time lag (such as seed back in this case)

• Increasing numbers of interacting species

• Asymmetric interactions

It would be impossible for the authors to tackle all of these ideas, but without alteast some exploration in the text, I don’t know that I see this as more than a nice example and perhaps not broad enough for the journal. I think this can be fairly easily remedied. The authors might feel the natural desire to defend their ideas and avoid being too critical, but I prefer that when a new modeling approach is proposed that we can assess both its strengths and weaknesses. The paper lacks clear guidance and when such a workflow is a good idea and when it would be problematic and rather opts for statements like L354

‘The fact that this two-step process enhances predictions over a one-step model with all data available is remarkable’

Comments

The authors are really taking on a large and important moment in quantitative community ecology. Decades of mechanistic models show relatively little promise in predicting population abundances dynamics in natural settings over broad spatial areas. These conditions need to made clear in the paper, there are plenty of very good community microcosm experiments (https://scholar.google.com/citations?hl=en&user=Nkgv64gAAAAJ&view_op=list_works&sortby=pubdate). See several papers by H. Lynch in predicting penguin population dynamics (particularly https://besjournals.onlinelibrary.wiley.com/doi/full/10.1111/1365-2656.12790) or very similar work by J. HillrisLambers (https://onlinelibrary.wiley.com/doi/abs/10.1111/ele.13236) showing that current models can’t produce large scale dynamics. I’ll avoid citing my own work here, but there are many similar overarching papers outlining the need for prediction in community ecology [1–3]. I don’t think the paper needs to repeat the work in those papers, but it needs to help set the stage. Rather than creating a simple old/new dichotomy between mechanistic and data-driven models (L24), I think the authors need to spend more time in setting up the tradeoffs between the two. Why were mechanistic models originally favored? Data availability, lack of computational power to do MCMC? Focus on hypothesis testing over prediction (see [4])? To make the kind of impact they want, the authors need to slow down in the intro, give some specific examples, rather than general citations, and really engage with the current state of the field. Try to avoid proposing a ‘solution’ (always dangerous, L22) rather quickly. I’m a very sympathetic reader, since I already use these kinds of models, but the target audience should be a much broader group of quantitative ecologists, so the arguments need to be well laid out.

L51. I work with these kinds of ML ecological models every day. I emphatically reject they have ‘solved’ anything (famously see [5]). I could show some well-constructed and thoughtful neural networks that do an absolutely terrible job at prediction compared to regression. Similar to the comment above, if the authors want to really make a contribution in this area, more subtly and accuracy is needed. L55 is particularly worrisome and shows some CS envy. This kind of sentence makes me stop reading a paper.

One thing that jumps out at me is that the virtue of the sequential ML models, especially in the computer vision space, has been their ability to scale to enormous amounts of data (going back to [6]). The argument has always been that the learning potential for these types of models greatly exceeds the formal hierarchical Bayesian models and does away with needing to set priors. But in ecology we almost never have such large datasets (though its getting easier). I worry about the applicability of this kind of method to the average ecology dataset, especially as the number of species increases. How would this scale? I don’t have an intuition for which parts of the model would be most sensitive to larger number of taxa. Similarly for spatial heterogeneity of the landscape. The authors should either dial back their language in the conclusion or look to simulations to help explore when these kind of models will be most successful.

Figure 2 B needs some text to help orient me. What do the authors hope to convey with this right-hand panel?

Minor Comments:

Slightly awkward phrasing in the abstract ‘inform back’. In general, I try not to provide too many line edits unless they interfere with readability. Missing ‘e’ in ‘especially’, in the abstract.

Try to be careful in the introduction to consider a wide range of sources. The topic is heated and will be best received if the authors don’t cite their own work in such prominent spots (e.g. L7-11).

1. Anderegg LDL, HilleRisLambers J. Local range boundaries vs. large-scale trade-offs: climatic and competitive constraints on tree growth. Ecol Lett. 2019;22: 787–796. doi:https://doi.org/10.1111/ele.13236

2. Youngflesh C, Jenouvrier S, Hinke JT, DuBois L, Leger JS, Trivelpiece WZ, et al. Rethinking “normal”: The role of stochasticity in the phenology of a synchronously breeding seabird. J Anim Ecol. 2018;87: 682–690. doi:https://doi.org/10.1111/1365-2656.12790

3. Dietze MC, Fox A, Beck-Johnson LM, Betancourt JL, Hooten MB, Jarnevich CS, et al. Iterative near-term ecological forecasting: Needs, opportunities, and challenges. Proc Natl Acad Sci. 2018;115: 1424–1432. doi:10.1073/pnas.1710231115

4. Betts MG, Hadley AS, Frey DW, Frey SJK, Gannon D, Harris SH, et al. When are hypotheses useful in ecology and evolution? Ecol Evol. n/a. doi:https://doi.org/10.1002/ece3.7365

5. Nguyen A, Yosinski J, Clune J. Deep Neural Networks Are Easily Fooled: High Confidence Predictions for Unrecognizable Images. 2015. pp. 427–436. Available: https://www.cv-foundation.org/openaccess/content_cvpr_2015/html/Nguyen_Deep_Neural_Networks_2015_CVPR_paper.html

6. Dean J, Corrado GS, Monga R, Chen K, Devin M, Le QV, et al. Large Scale Distributed Deep Networks. NIPS. 2012.

Reviewer #3: Kia ora koutou,

Thank you so much for offering me the opportunity to read carefully your manuscript.

The manuscript is clear, and the methodology interesting and robust.

There may be some minor revision needed to convey the message more smoothly and I indicate them in the attached PDF document.

Best wishes,

**Have the authors made all data and (if applicable) computational code underlying the findings in their manuscript fully available?**

Reviewer #1: Yes

Reviewer #2: Yes

Reviewer #3: Yes

PLOS authors have the option to publish the peer review history of their article (what does this mean?). If published, this will include your full peer review and any attached files.

Reviewer #1: No

Reviewer #2: No

Reviewer #3: **Yes: **Giulio Valentino Dalla Riva
---

## [Decision Letter · Decision Letter 1]

27 Sep 2021

Dear Dr García-Algarra,

Thank you very much for submitting your manuscript "Fine scale prediction of ecological community composition using a two-step sequential machine learning ensemble" for consideration at PLOS Computational Biology. As with all papers reviewed by the journal, your manuscript was reviewed by members of the editorial board and by several independent reviewers. The reviewers appreciated the attention to an important topic. Based on the reviews, we are likely to accept this manuscript for publication, providing that you modify the manuscript according to the review recommendations.

Sincerely,

Jacopo Grilli

Associate Editor

PLOS Computational Biology

Natalia Komarova

Deputy Editor

PLOS Computational Biology

[LINK]

Reviewer's Responses to Questions

**Comments to the Authors:**

Reviewer #1: First, I want to thank the authors for taking most of my prior suggestions seriously and working hard to improve their manuscript. In the comments below I’ll focus on specific (new) line numbers where issues remain, as well as the comment numbers the authors introduced in their response to my review.

Introduction: At a high-level, I’m still concerned that the authors are couching their paper in terms of the (real) scaling problems inherent in theory-based modeling approaches to species interactions (e.g. Lotka-Volterra; Lines 21, 25-48, etc), but then glosses over that (A) the method they propose has the exact same issue (number of terms scales quadratically with the number of species and (B) other mechanistic modeling frameworks exist that don’t have this scaling issue. See previous Comments XX.

Line 56: The authors point to the lack of interpretability of machine learning models, but then go on to present an uninterpretable analysis. For example, there are still no plots showing the shapes of the relationships inferred (previous Comment XX). The two-stage modeling approach is also particularly hard to interpret because it’s not clear what these stages represent. Counter to what is implied (but not outright stated, e.g. Line 71 and 406), the ABIOTIC model is most definitely NOT a model of the fundamental niche, but instead is a model of the realized niche, with the impacts of species interactions still implicitly present in the data.

Equation 2: It’s not clear to me what point is being made here. The only difference between equation 1 and 2 is that you’re calling the function g instead of f.

Equation 3: As noted in previous Comment XX, I have real concerns about the approach to temporal that’s implied here, which differs strongly from standard time-series and population modeling approaches (which is to use X_t to predict X_t+1). Indeed, in this round of review I downloaded the authors data and can confirm that, counter to their abysmal results in Table 3, a linear model that just use X_t to predict X_t+1 (without ANY other covariates) has an R2 of 0.42 (compared to the reported -3.95 to 0.08). Adding in species identity and the precip for both time t and t+1 (but not ANY soil variables, species interactions, interactions between species and precip, or nonlinear relationships), a simple linear model increases to a R2 of 0.56. Admittedly this is still worse than the spatial Random Forest, but it does strongly suggest that the Authors revise their approach and all of their Results and Discussion around temporal models.

As a separate comment, accessing the raw data made it clear to me that the soil covariate data for a subplot were identical for every single year. First, this was not obvious to me when reading the paper so should be more prominent in the Methods. Second, it should be reiterated in the Results and Discussion, as it’s utterly unsurprising that their current Temporal model does so poorly when the covariate data used to predict Temporal variations is itself not changing in time.

Line 118: measuring, not measure

Line 134: Unless you have a time machine I don’t know about, measuring abiotic data for time t+1 when you’re currently at time t isn’t possible. If you want to predict to time t+1 you need to either use covariates at time t or (uncertain) forecasts of covariates to time t+1. Second point, throughout the paper (here, L147, etc.), the authors keep saying that it’s easier to measure abiotic covariates than it is to measure the biotic data itself (species composition) but they don’t provide any evidence to support this argument. Indeed, as previously noted their own data set samples species every year, but abiotic covariates only once, strongly implying that the latter is harder. On top of this, there are known technological approaches to measuring species composition at fine spatial resolutions and non-trivial spatial scales (e.g. NEON’s 1 m^2 resolution imaging spectroscopy), but as far as I know there is no equivalent technological approach to measuring below-ground nutrient concentrations.

Line 150: I’m not sure why the authors think this approach is “too radical” since they’re just describing species distribution modeling, which is widespread in the discipline.

Line 166: Unclear how the uncertainties in the predicted values are being handled in the second stage model. Please comment on this explicitly. I suspect they are being ignored, in which case this should be revisited in the Discussion as a limitation / future direction.

L198: still not adequately addressing how the approached described deal with the zero inflation problem (previous Comment 2)

Line 304-305: sentence very hard to parse. I had to read it 3-4 times just to be sure it was a sentence, not a fragment, and it was still unclear what the authors were trying to convey.

Line 311: Not entirely following why this was done over just not including species j as a covariate when predicting species j.

Line 322: latter not former

Figure 3: The decision to both use a different species order in panel B, and to switch the mapping between color and species, is extremely misleading. I think it is essential to keep the colors the same between panels and recommend keeping the species order the same too.

Line 356: why? Please revisit in Discussion

Line 369: First, this comes as a surprise to me as I would have expected a simple repeated measures model to have done fairly well because of the tendency of plants to stay put through time. Second, as discussed above I don’t think this analysis is correct.

L377: ambiguous what “this variable” is referring to.

L391: Feature

Line 392: Still introducing new Methods in the Results. This is not OK. Per previous Comment 8, you really need to lay out this analysis in the Methods and preferably also motivate it in the Introduction.

Line 444: This likewise feels like new Methods and Results in the Discussion. Mechanistic modeling part not adequately explained.

Line 476: NEON is all caps, and acronyms should be defined, and it needs a citation

Comment 5: I’ll defer to the Editor, but I’m not fully convinced that a comparison to conventional methods for vegetation analysis (e.g. CCA) is “beyond the scope”. Seems like at a minimum the existence of such methods should come up in the Discussion.

Comment 6: did not address the issue about differences in sensitivity to spatial and temporal anomalies. Would be good to add a quick line to the Discussion.

Reviewer #2: I appreciate the authors willingness to engage with reviewers and improve the text. The intro is much better. I like the idea and think it is worth exploring. The authors continue to overstate the strength of their results, but I think this point is stylistic and rather minor. A good revision.

Reviewer #3: Kia ora koutou

Thank you so much for you effort in answering my questions.

I'm satisfied with the answer and the edits.

Ngā mihi,

Giulio

**Have the authors made all data and (if applicable) computational code underlying the findings in their manuscript fully available?**

Reviewer #1: Yes

Reviewer #2: Yes

Reviewer #3: Yes

PLOS authors have the option to publish the peer review history of their article (what does this mean?). If published, this will include your full peer review and any attached files.

Reviewer #1: No

Reviewer #2: **Yes: **Ben Weinstein

Reviewer #3: **Yes: **Giulio Valentino Dalla Riva

Figure Files:

Data Requirements:

Reproducibility:

References:

---

## [Editor Report · Decision Letter 2]

12 Nov 2021

Dear Dr García-Algarra,

We are pleased to inform you that your manuscript 'Fine scale prediction of ecological community composition using a two-step sequential machine learning ensemble' has been provisionally accepted for publication in PLOS Computational Biology.

Best regards,

Jacopo Grilli

Associate Editor

PLOS Computational Biology

Natalia Komarova

Deputy Editor

PLOS Computational Biology

---

## [Editor Report · Acceptance letter]

1 Dec 2021

PCOMPBIOL-D-21-00507R2 

Fine scale prediction of ecological community composition using a two-step sequential machine learning ensemble

Dear Dr García-Algarra,

I am pleased to inform you that your manuscript has been formally accepted for publication in PLOS Computational Biology. Your manuscript is now with our production department and you will be notified of the publication date in due course.

With kind regards,

Olena Szabo
